# Anticancer Effect of Hemin through ANO1 Inhibition in Human Prostate Cancer Cells

**DOI:** 10.3390/ijms25116032

**Published:** 2024-05-30

**Authors:** So-Hyeon Park, Yechan Lee, Hyejin Jeon, Junghwan Park, Jieun Kim, Mincheol Kang, Wan Namkung

**Affiliations:** 1College of Pharmacy and Yonsei Institute of Pharmaceutical Sciences, Yonsei University, 85 Songdogwahak-ro, Yeonsu-gu, Incheon 21983, Republic of Korea; sohyeon0605@yonsei.ac.kr (S.-H.P.); llyycc94@naver.com (Y.L.); isy0803@naver.com (H.J.); penseroso97@gmail.com (J.P.); 2Graduate Program of Industrial Pharmaceutical Science, Yonsei University, 85 Songdogwahak-ro, Yeonsu-gu, Incheon 21983, Republic of Korea; jieunkimx@yonsei.ac.kr (J.K.); alscjf1592@naver.com (M.K.)

**Keywords:** hemin, anoctamin 1, inhibitor, prostate cancer, PC-3

## Abstract

Anoctamin1 (ANO1), a calcium-activated chloride channel, is overexpressed in a variety of cancer cells, including prostate cancer, and is involved in cancer cell proliferation, migration, and invasion. Inhibition of ANO1 in these cancer cells exhibits anticancer effects. In this study, we conducted a screening to identify novel ANO1 inhibitors with anticancer effects using PC-3 human prostate carcinoma cells. Screening of 2978 approved and investigational drugs revealed that hemin is a novel ANO1 inhibitor with an IC_50_ value of 0.45 μM. Notably, hemin had no significant effect on intracellular calcium signaling and cystic fibrosis transmembrane conductance regulator (CFTR), a cyclic AMP (cAMP)-regulated chloride channel, and it showed a weak inhibitory effect on ANO2 at 3 μM, a concentration that completely inhibits ANO1. Interestingly, hemin also significantly decreased ANO1 protein levels and strongly inhibited the cell proliferation and migration of PC-3 cells in an ANO1-dependent manner. Furthermore, it strongly induced caspase-3 activation, PARP degradation, and apoptosis in PC-3 cells. These findings suggest that hemin possesses anticancer properties via ANO1 inhibition and could be considered for development as a novel treatment for prostate cancer.

## 1. Introduction

ANO1 (anoctamin1), also called transmembrane protein 16A (TMEM16A), is a calcium-activated chloride channel (CaCC) [1,2,3]. It is widely expressed throughout the body, including in smooth muscle, salivary gland epithelial cells, and sensory neurons, where it regulates a variety of physiological responses [4,5,6]. ANO1 is located on chromosome 11q13, which is frequently amplified in malignant tumors [7,8], and is reported to be highly expressed in glioblastoma [9], head and neck squamous cell carcinoma [10], gastrointestinal cancer [11], pancreatic cancer [12], esophageal cancer [13], breast cancer [14], and prostate cancer [15]. This high expression of ANO1 protein is associated with poor prognostic survival in cancer patients. Although the underlying mechanism is still unclear, inhibition and downregulation of ANO1 have been shown to inhibit cell migration, proliferation, and invasion in cancer cells highly expressing ANO1. For instance, ANO1 knockdown with small hairpin RNAs (shRNAs) inhibited the migration, proliferation, and invasion of human lung cancer cell, and silencing ANO1 in vivo reduced tumor growth [16]. Furthermore, strong expression of ANO1 in PC-3 cells and its decrease after shRNA treatment were observed, and downregulation of ANO1 expression by intratumoral injection of ANO1 shRNA in a xenograft mouse model of prostate cancer using PC-3 cells significantly inhibited tumor growth [17].

Prostate cancer is one of the most prevalent cancers, ranking as the second leading cause of cancer-related deaths in men, following lung cancer [18]. The burden of prostate cancer is anticipated to increase with the aging of the population [19]. Current treatment options for prostate cancer include chemotherapy, hormone therapy, radiation therapy, cryotherapy, and surgery. However, these treatments are associated with serious side effects such as toxicity, decreased white and red blood cell counts, hair loss, peripheral neuropathy, erectile incontinence and dysfunction, metastasis, and the development of resistance. Moreover, these treatment options are often costly and associated with severe side effects, highlighting the need for new cost-effective chemotherapeutic agents with low side effects and high efficacy. Interestingly, recent studies revealed that ANO1 is highly expressed in prostate cancer, and the inhibition of ANO1 demonstrated anticancer effects [15,17]. Therefore, ANO1 may be a promising therapeutic target for prostate cancer.

Several ANO1 inhibitors have been identified so far, such as CaCC_inh_-A01, tannic acid, T16A_inh_-A01, MONNA, idebenone, Ani9, Ani9-5f, Ani-D2, and luteolin [20,21,22,23,24,25,26]. In particular, Ani9, which we previously identified, exhibited submicromolar potency in inhibiting the activity of the ANO1 channel, surpassing T16A_inh_-A01 and MONNA, and demonstrated high selectivity compared to ANO2 [24]. Furthermore, in our previous studies, we have shown that luteolin, Ani9-5f, and Ani-D2 reduce cell proliferation and induce apoptosis in PC-3 cells expressing high levels of ANO1 through downregulation of ANO1 [23,25,26]. However, previously identified ANO1 inhibitors are still in the preclinical study stage as it takes a long time to develop ANO1 inhibitors suitable for clinical application.

In the present study, we performed a cell-based screening for the identification of ANO1 inhibitor and found a novel ANO1 inhibitor, hemin. Hemin, an iron-containing porphyrin, has been commonly utilized for the treatment of acute porphyria [27]. Here, we evaluated the effect of hemin, which has high clinical applicability, on ANO1 and its anticancer effects on prostate cancer cells.

## 2. Results

### 2.1. Identification of a Novel ANO1 Inhibitor

We performed a cell-based screening of a chemical library containing 2978 approved and investigational drugs to identify novel ANO1 inhibitors using the YFP fluorescence quenching assay. The functional activity of ANO1 chloride channel was evaluated via stable transfection of iodide-sensing mutant YFP (F46L/H148Q/I152L) into PC-3 cells with high endogenous ANO1 expression. Eleven compounds that blocked more than 80% of iodide influx through ANO1 were identified as hits, including previously known ANO1 inhibitors such as benzbromarone, dichlorophen, sanguinarine chloride, and diethylstilbestrol [22,28]. Notably, hemin emerged as the only potent new inhibitor showing selectivity for ANO1. The chemical structure of hemin is shown in Figure 1A. In PC-3 cells, hemin potently inhibited ANO1 channel activity with IC_50_ value of 0.45 μM (Figure 1B,C).

To investigate more clearly whether hemin inhibits iodide influx through direct inhibition of ANO1, a YFP fluorescence quenching assay was conducted in Fischer rat thyroid (FRT) cells stably expressing human ANO1. As shown in Figure 1D,E, hemin potently inhibited ANO1 channel activity with an IC_50_ value of 0.51 μM. To investigate the impact of hemin on ANO1 chloride current, short-circuit current measurements were conducted in FRT-ANO1 cells. Hemin exhibited a potent and dose-dependent inhibition of ATP-induced ANO1 chloride current (Figure 1F).

### 2.2. Characterization and Selectivity of a Novel ANO1 Inhibitor, Hemin

To investigate whether hemin affects intracellular calcium levels, PC-3 cells were loaded with Fluo-4, a calcium indicator, and then the effect of hemin on ATP-induced intracellular calcium increase was observed. Hemin had minimal effect on ATP-induced intracellular calcium increase up to 3 μM, a concentration that completely inhibits ANO1, and it weakly inhibited ATP-induced intracellular calcium increase at 10 μM (Figure 2A). To observe the effect of hemin on other chloride channels, ANO2 and cystic fibrosis transmembrane conductance regulator (CFTR), we analyzed the effect of hemin on ANO2 and CFTR in FRT cells expressing human ANO2 and CFTR, respectively. To determine whether hemin inhibits ANO2 channel activity, we performed a YFP fluorescence quenching assay in FRT-ANO2-YFP cells. As shown in Figure 2B, hemin strongly inhibited ANO2 channel activity at 10 μM, but had a little effect on ANO2 at a concentration of 3 μM. The IC_50_ values for ANO1 and ANO2 were 0.51 μM and 4.09 μM, respectively (Figure 1E and Figure 2C). To investigate the effect of hemin on CFTR, the apical membrane currents were measured in FRT-CFTR cells. Hemin did not alter CFTR chloride channel activity up to 10 μM (Figure 2D).

### 2.3. Effect of Hemin on Protein and mRNA Expression Levels of ANO1

To observe the effect of hemin on ANO1 protein expression levels, Western blot analysis was performed in PC-3 cells. Remarkably, hemin reduced ANO1 protein expression levels in a dose-dependent manner (Figure 3A,B). Real-time PCR analysis was performed to investigate whether hemin affects ANO1 mRNA expression levels. As shown in Figure 3C, hemin did not alter the mRNA expression levels of ANO1.

### 2.4. Inhibitory Effects of Hemin on Cell Proliferation and Migration in PC-3 Cells

Previous studies have shown that inhibition of ANO1 decreases cell proliferation and migration in prostate cancer cells [22,25,26]. To investigate whether hemin exerts an ANO1-dependent effect on cell growth, the effect of hemin on cell viability was examined in other cancer cells that do not express ANO1. In addition, we generated a PC-3 cell line with ANO1 knockout (KO) using CRISPR-Cas9. As shown in Figure 4A, PC-3 ANO1 KO, LNCaP, A549, and HepG2 cells demonstrated undetectable expression of ANO1, in contrast to PC-3 cells exhibiting high ANO1 expression. Interestingly, the reduction in cell viability induced by hemin was significantly higher in PC-3 cells compared to PC-3 ANO1 KO cells (Figure 4B). Furthermore, at 10 μM hemin, ANO1 KO PC-3 cells exhibited a slightly significant decrease in cell viability to 13.36%, whereas PC-3 cells showed a highly significant reduction to 45.74%.

To demonstrate the specific impact of ANO1 on prostate cancer cells, we evaluated cell viability in PC-3 ANO1 KO cells transiently subjected to ANO1 transfection (Appendix A). Notably, treatment with 10 μM hemin significantly reduced cell viability by 29.28% in the ANO1 rescue cells compared to control (Figure 4C). In contrast, hemin has shown minimal impact on the cell viability of LNCaP (prostate cancer), A549 (lung cancer), and HepG2 (liver cancer) cells, which express negligible levels of ANO1, even at high concentrations (Figure 4D–F). These results suggest that hemin specifically reduces cell viability in cancer cells highly expressing ANO1.

To observe whether hemin inhibits cell migration in an ANO1-dependent manner, we performed wound healing assays in PC-3 and PC-3 ANO1 KO cells. Interestingly, hemin significantly reduced cell migration in a dose-dependent manner in PC-3 cells (Figure 5A,B). However, hemin had no significant effect on cell migration of PC-3 ANO1 KO cells, which do not express ANO1 (Figure 5C,D).

### 2.5. Hemin Induces Apoptosis and Cell Cycle Arrest in PC-3 Cells

Previous studies have indicated that inhibition of ANO1 leads to apoptosis in prostate cancer cells [29]. To determine whether hemin induces apoptosis in PC-3 cells highly expressing ANO1, we observed the effect of hemin on PARP cleavage and caspase-3 activity which are considered as hallmarks of apoptosis. As shown in Figure 6, hemin significantly increased PARP cleavage in a dose-dependent manner in PC-3 cells (Figure 6A,B). In addition, hemin significantly increased caspase-3 activity compared to control, and the hemin-induced increase in caspase-3 activity was almost completely blocked by Ac-DEVD-CHO, a caspase-3 inhibitor (Figure 6C,D).

To investigate the effect of hemin on the cell cycle of PC-3 cells, flow cytometry analysis using propidium iodide (PI) was performed. Since high concentration of hemin exhibited potential off-target effects such as intracellular calcium levels and ANO2 activity (Figure 2A–C), we selected a concentration of 3 µM hemin for the flow cytometry analysis, at which hemin almost completely inhibited ANO1 while minimizing off-target effects on other targets. Notably, hemin significantly increased the sub-G1 (apoptotic peak) ratio from 14.18% to 29.32% in 48 h compared to the control (Figure 7). In addition, hemin induced a decrease in the G0/G1 phase from 77.32% to 59.20% and an increase in the G2/M phase from 3.85% to 6.13%.

## 3. Discussion

Hemin, an oxidized form of heme, is abundant in human erythrocytes, with approximately 2.5 mM hemoglobin in the blood and 10 mM hemin after degradation [30]. Hemin is currently used in the treatment of acute intermittent porphyria (AIP), a rare inherited metabolic disorder characterized by a deficiency in porphobilinogen deaminase, which disrupts heme biosynthesis [31]. AIP is treated by inhibiting porphyrin production through hemin injections. Given that hemin has been safely administered as a daily infusion at doses ranging from 1 to 4 mg/kg for 3 to 14 days in the treatment of porphyria [32], it holds significant promise for drug development. Therefore, in the present study, we investigated the ANO1-dependent anticancer effects of hemin, a novel ANO1 inhibitor, in prostate cancer cells.

Recent evidence suggests that ANO1 is a potential therapeutic target for various cancers, including gastrointestinal stromal cancer (GIST), head and neck squamous cell carcinoma (HNSCC), and oral squamous cell carcinoma (OSCC) [10,11,33]. Specifically, ANO1 has been reported as a new drug target for prostate cancer [15,23,24,25,26]. Interestingly, hemin, a novel ANO1 inhibitor, showed anticancer effects dependent on ANO1 expression in PC-3 prostate cancer cells. As shown in Figure 4 and Figure 5, hemin significantly increased cytotoxicity and decreased cell migration in a concentration-dependent manner in PC-3 cells highly expressing ANO1. Moreover, hemin significantly promoted caspase 3 activation and PARP cleavage and induced cell cycle arrest, as demonstrated by an increased the apoptotic sub-G1 peak and G2/M population along with a decreased G0/G1 population in PC-3 cells (Figure 6 and Figure 7). However, in PC-3 ANO1 KO cells, hemin did not have a significant effect on cytotoxicity and cell migration. Notably, unlike hemin, ANO1 inhibitors such as luteolin and Ani-D2 showed nonspecific cytotoxic effects in PC-3 ANO1 KO cells [25,26]. In addition, hemin showed little cytotoxicity in LNCaP, A549, and HepG2 cells that barely express ANO1 (Figure 4) and did not significantly affect the CFTR chloride channel activity and intracellular calcium even at high concentrations (Figure 2A,D). ANO2 was significantly blocked by hemin in a concentration-dependent manner (Figure 2B,C). However, hemin has approximately eight times higher selectivity for ANO1 compared to ANO2 in terms of the IC_50_ values. Since hemin is widely used in clinics for the treatment of porphyria; it has high potential to be developed as a treatment for prostate cancer that overexpresses ANO1.

Hemin is known to regulate the expression of cancer signal transduction proteins and enzymes involved in lipid metabolism [34]. Interestingly, hemin has been reported to have anticancer activity in several cancer cells, including prostate, breast, lung, and colon cancer cells [35,36,37,38,39]. For example, combined treatment of hemin and ionizing radiation increased ferroptosis radiosensitivity of lung cancer cells through promoting lipid peroxidation, ROS production, and GPX4 degradation. Meanwhile, hemin protected normal lung cells from ionizing radiation by increasing bilirubin levels and shields iron by enhancing FTH1 expression [37]. Hemin is a representative heme oxygenase-1 (HO-1) inducer. HO-1 is an enzyme that catalyzes the oxidative degradation of heme to liberate free iron, carbon monoxide, and biliverdin and is highly induced in several cancer [40]. A previous study showed that endogenous HO-1 inhibits the proliferation, invasion, and migration of bone-derived prostate cancer cells and reduces tumor growth and angiogenesis in vivo [41]. Furthermore, Gueron, G. et al. also examined the effect of hemin on toxicity and HO-1 expression in PC-3 cells, and they found that hemin exhibited a cytotoxic effect and significantly enhanced HO-1 expression at concentrations ranging from 30 µM to 80 µM [41]. When we observed the effect of high concentrations of hemin on cell viability in PC-3 and PC-3 ANO1 KO cells, as expected, there was no significant difference in cell viability between PC-3 and PC-3 ANO1 KO cells treated with 30 μM of hemin, which decreased to 49% and 40%, respectively (Appendix A). Notably, here we observed no cytotoxicity of hemin in ANO1 KO PC-3 cells up to 3 µM, but hemin induced significant cytotoxicity and apoptosis in PC-3 cells highly expressing ANO1 (Figure 4B and Figure 6). Therefore, these results suggest that in PC-3 prostate cancer cells, low concentrations of hemin may exhibit ANO1-dependent anticancer effects, and high concentrations of hemin may exhibit additional anticancer effects via other pathways such as increasing HO-1 expression. Notably, HO-1 plays a dual role in prostate cancer, and its effects vary depending on the context and level of its expression, where major increases in ROS lead to protective effects in normal cells, while moderate increases promote cancer progression [42]. Therefore, both inhibiting and inducing HO-1 can demonstrate anticancer effects in prostate cancer models. Consequently, additional investigations are warranted to elucidate the effect of hemin on HO-1 activity and its potential implications for anticancer strategies.

Previous studies have endeavored to elucidate the pathophysiological role of ANO1 in prostate cancer. Both ANO1 inhibitors and downregulation of ANO1 have exhibited anticancer effects against prostate cancer [15,17]. In a previous study, we showed that luteolin and its structural analog kaempferol exhibit comparable potencies in inhibiting the activity of the ANO1 chloride channel. However, luteolin, which strongly reduces ANO1 protein expression, inhibited prostate cancer cell proliferation more effectively than kaempferol, which does not [26]. These findings suggest that modulation of ANO1 protein expression levels may exert a more profound impact on cell proliferation compared to mere inhibition of its channel activity. Downregulation of ANO1 can not only abrogate its channel functions but also disrupt ANO1-mediated proliferative signaling cascades, such as the potentiation of EGFR signaling pathways [14]. Herein, we demonstrate that hemin inhibits ANO1 channel activity while concomitantly reducing ANO1 protein expression levels (Figure 1B and Figure 3A). Therefore, we hypothesize that hemin may elicit potent anticancer effects in prostate cancer cells through the dual mechanisms of ANO1 channel blockade and ANO1 downregulation.

In conclusion, hemin potently and selectively inhibited ANO1 channel activity and decreased the protein expression level of ANO1 without alteration of ANO1 mRNA expression in PC-3 cells. Remarkably, hemin exhibited ANO1-dependent inhibition of cell growth and migration and induced of apoptosis in PC-3 cells while affecting cell cycle arrest. Taken together, these results suggest that hemin, currently used clinically, has a potential to be developed as a new therapeutic agent for prostate cancer overexpressing ANO1.

## 4. Materials and Methods

### 4.1. Cell Culture and Cell Lines

FRT (Fischer rat thyroid) cell lines stably expressing ANO1, ANO2, or CFTR with a YFP variant (F46L/H148Q/I152L) were provided by Alan Verkman (University of California, San Francisco, CA, USA) and cultured in Dulbecco’s Modified Eagle’s Medium/F-12 modified Coon’s medium with 10% fetal bovine serum (FBS), 100 units/mL penicillin, 100 μg/mL streptomycin, and 2 mM L-glutamine at 37 °C with 5% CO_2_. PC-3, LNCaP, and A549 cells were cultured in RPMI-1640 medium, while HepG2 cells were cultured in DMEM medium supplemented with 10% FBS, 100 units/mL penicillin, and 100 μg/mL streptomycin at 37 °C with 5% CO_2_. PC-3, LNCaP, A549, and HT29 cells were purchased from Korean Cell line Bank (Seoul, Republic of Korea).

### 4.2. Materials and Reagents

A chemical library containing 2978 approved and investigational drugs (TargetMol, Boston, MA, USA) was used for cell-based screening. The compounds in the library were dissolved in dimethyl sulfoxide solution (DMSO), and cells were treated with a final concentration of 1% DMSO. Hemin, ATP, CFTR_inh_-172, CaCC_inh_-A01, cisplatin and AC-DEVD-CHO were purchased from Sigma-Aldrich (St. Louis, MO, USA). Propidium iodide (PI) were purchased from Invitrogen (Carlsbad, CA, USA). All modulators were dissolved in DMSO, which was also used as the vehicle control. All cells were treated with 0.1% (*v*/*v*) DMSO as the final concentration, except for the cell viability assay treated with 0.5% (*v*/*v*) DMSO.

### 4.3. YFP Fluorescence Quenching Assay

PC-3, FRT-ANO1, and FRT-ANO2 cells stably expressing YFP variant (F46L/H148Q/I152L) were plated in 96-well microplates at a density of 2 × 10^4^ cells per well. After 48 h incubation, each well of the 96-well plates were washed twice in phosphate-buffered solution (PBS, 200 μL/wash), and test compounds (10 μM) were added to each well. After 10 min incubation at 37 °C, the 96-well plates were transferred to a FLUOstar Omega microplate reader (BMG Labtech, Ortenberg, Germany) for YFP fluorescence assay. Iodide solution containing 100 μM of ATP was added at 1 s (baseline) using a syringe pump, and each well iodide flux was monitored for 5 s and continuously recorded 400 ms per point. The inhibition effect of ANO1 channel activity was measured according to the initial slope of YFP fluorescence decrease.

### 4.4. Cytoplasmic Calcium Measurements

PC-3 cells were cultured in 96-well microplates at a density of 2 × 10^4^ cells per well. After 48 h incubation, the cells loaded with Fluo-4 NW calcium assay kit (Invitrogen, Carlsbad, CA, USA) following the manufacturer’s protocol. Briefly, the cells were incubated with 100 μL assay buffer containing Fluo-4. After 40 min of incubation, cells were treated with compounds for 10 min and the 96-well plates were transferred to a FLUOstar Omega microplate reader (BMG Labtech, Ortenberg, Germany) for fluorescence assay. Fluo-4 fluorescence was equipped with syringe pumps and Fluo-4 excitation/emission filters (485/538 nm). Intracellular calcium signaling was increased by application of 100 μM ATP.

### 4.5. Short-Circuit Current

FRT cells stably expressing ANO1 and CFTR were seeded at a confluence of 2 × 10^5^ cells/cm^2^ on Snapwell inserts (1.12 cm^2^ surface area) and cultured until confluent. Snapwell inserts were mounted in Using chambers (Physiologic Instruments, San Diego, CA, USA). The basolateral bath was stocked with HCO_3_^−^ buffered solution containing 120 mM NaCl, 5 mM KCl, 1 mM MgCl_2_, 1 mM CaCl_2_, 10 mM D-glucose, 2.5 mM HEPES, and 25 mM NaHCO_3_ (pH 7.4). The apical was bathed with half-Cl^-^ solution (70 mM NaCl in the HCO_3_^−^-buffered solution was replaced with Na-gluconate). Cells were incubated for 30 min and aerated with 95% O_2_, 5% CO_2_, at 37 °C. Hemin was pretreated with apical and basolateral bath solutions 10 min before ANO1 activation. After 10 min, 100 µM of ATP or 10 µM forskolin was treated to the apical bathing solution, and then apical membrane currents were measured with a Power Lab 4/35 (AD Instruments, Castle Hill, Australia) and EVC4000Multi-Channel V/I Clamp (World Precision Instruments, Sarasota, FL, USA). Data were evaluated using Lab chart Pro 7 (AD Instruments, Castle Hill, Australia). The sampling rate was set at 4 Hz.

### 4.6. Immunoblotting

Cells were lysed in cell lysis buffer (50 mM Tris-HCl (pH 7.4), 150 mM NaCl, 1% Nonidet P-40, 0.25% sodium deoxycholate, 1 mM EDTA, 1 mM Na_3_VO_4_, protease inhibitor mixture). Whole-cell lysates were centrifuged at 13,000 rpm for 20 min at 4 °C, and supernatant proteins of equal amounts (80 μg protein/lane) were separated via 4–12% Tris–glycine precast gel (KOMA BIOTECH, Seoul, Republic of Korea). Then, separated proteins were transferred onto a PVDF membrane (Millipore, Billerica, MA, USA) and blocked with 5% non-fat skin milk in Tris-buffered saline (50 mM Tris-Cl, pH 7.5, 150 mM NaCl) with 0.1% Tween 20 (TBST) for 1 h at room temperature. The membranes were incubated overnight at 4 °C with primary antibodies, containing anti-TMEM16A antibody [SP31] (Abcam, Cambridge, UK), anti-cleaved PARP (BD Biosciences, Franklin Lakes, NJ, USA), and anti β-actin (Santa Cruz Biotechnology, Dallas, TX, USA). After washing with TBST 3 times at 5 min internals, membranes were incubated for 1 h at room temperature with HRP-conjugated anti-secondary IgG antibodies (Santa Cruz Biotechnology, Dallas, TX, USA). Visualization was conducted with the SuperSignal^TM^ Western Blot Substrate (Thermo Fisher Scientific, Waltham, MA, USA). All experiments were repeated three times independently, and ImageJ software version 1.53t (NIH, Bethesda, MD, USA) was used for result analysis.

### 4.7. Real-Time RT-PCR Analysis

Total mRNA was extracted using TRIzol solution (Invitrogen, Carlsbad, CA, USA). Total mRNA was reverse transcribed with hexamer primers, an oligo (dT) primer, and SuperScript III Reverse Transcriptase (Invitrogen, Carlsbad, CA, USA). StepOnePlus^TM^ Real-Time PCR program version 2.3 (Applied Biosystems, Waltham, MA, USA) and Thunderbird SYBR qPCR mix (Toyobo, Osaka, Japan) were used for quantitative real-time PCRs. The thermal cycling conditions consisted of an initial step at 95 °C for 5 min followed by 40 cycles of 95 °C for 10 s, 60 °C for 20 s, and 72 °C for 10 s. The ANO1 sense primer sequence was 5′-GGAGAAGCAGCATCTATTTG-3′; the ANO1 antisense primer sequence was 5′-GATCTCATAGACAATCGTGC-3′. The size of the ANO1 PCR product is 82 base pairs.

### 4.8. Cell Viability Assay

Cell Titer 96^®^ Aqueous One Solution Assay kit (Promega, Madison, WI, USA) was used to perform cell viability assay. Cells were cultured in 96-well plates with growth medium supplemented with 2% FBS for 24 h. When cell density reached ~30%, DMSO and hemin were treated in medium, exchanged freshly every 24 h. After 72 h incubation, the medium was washed out and the MTS assay was performed according to the supplier’s instructions. The absorbance at 490 nm was measured with an Infinite M200 microplate reader (Tecan, Männedorf, Switzerland).

### 4.9. Wound Healing Assay

PC-3 cells were seeded in 96-well plates for 24 h. When cell density reached approximately 100% confluency to form a monolayer, the medium was replaced with growth medium supplemented with 2% FBS. Wounds were inflicted using a 96-well Wound Maker (Essen Bio-Science, Ann Arbor, MI, USA). The cells were washed out twice with PBS and incubated with a medium containing the indicated concentrations of hemin or DMSO for 36 h. Images were acquired using IncuCyte ZOOM (Essen BioScience, Ann Arbor, MI, USA). The percentage of wound closure were calculated using IncuCyte software 2018A.

### 4.10. Caspase-3 Activity Assay

PC-3 cells were cultured in 96-well black plates at a density of 2 × 10^4^ cells per well, containing growth medium supplemented with 2% FBS for 24 h. Then, hemin and DMSO were treated in each well. After incubation for 24 h, each well was washed out twice with PBS and replaced with 100 μL of PBS containing 1 µM of caspase-3 substrate, NucView 488 (Biotium, Fremont, CA, USA), for 30 min at room temperature. Then, 1 µM of Hoechst 33342 (Thermo Fisher, Waltham, MA, USA) was added to the cells for 10 min. The activity of caspase-3 was inhibited by Ac-DEVD-CHO, a selective caspase-3 inhibitor. The fluorescence of NucView 488 was measured using a FLUOstar Omega microplate reader (BMG Labtech, Ortenberg, Germany), and fluorescence microscopy images of NucView 488 and Hoechst 33342 were captured with a Lionheart FX Automated Microscope (BioTek, Winooski, VT, USA).

### 4.11. Cell Cycle Analysis

PC-3 cells were seeded in 6-well plates at a density of 6 × 10^5^ cells per well. The medium containing hemin was replaced every 24 h. Cell cycle analysis was measured using flow cytometry following a standard protocol with some modifications [43]. Briefly, cells were washed out twice with PBS and trypsinized using 0.5% trypsin-EDTA. The harvest cells were centrifuged at 1000 rpm for 2 min at room temperature. The supernatant was discarded, and the pellet containing cells were resuspended in 100 μL of buffer solution (10 mM HEPES/NaOH, pH 7.4, 140 mM NaCl, 2.5 mM CaCl_2_) containing 2 μg/mL propidium iodide (PI). The cells were incubated in a dark room for 30 min at room temperature and added 500 μL cold buffer solution. Cell cycle analysis was performed using a flow cytometer (Beckman Coulter, Fullerton, CA, USA) and CytExpert software, version 2.4.0.28 (Beckman Coulter). For each measurement, at least 10,000 cells were counted.

### 4.12. Statistical Analysis

The results of all experiments were collected independently a minimum of three times and are presented as the mean ± standard deviation (S.D.). Statistical analysis was performed using Student’s *t*-test or one-way analysis of variance (ANOVA), as appropriate, using GraphPad Prism 5.0 (GraphPad Software Inc., San Diego, CA, USA). A value of *p* < 0.05 was considered statistically significant.

## Figures and Tables

**Figure 1 ijms-25-06032-f001:**
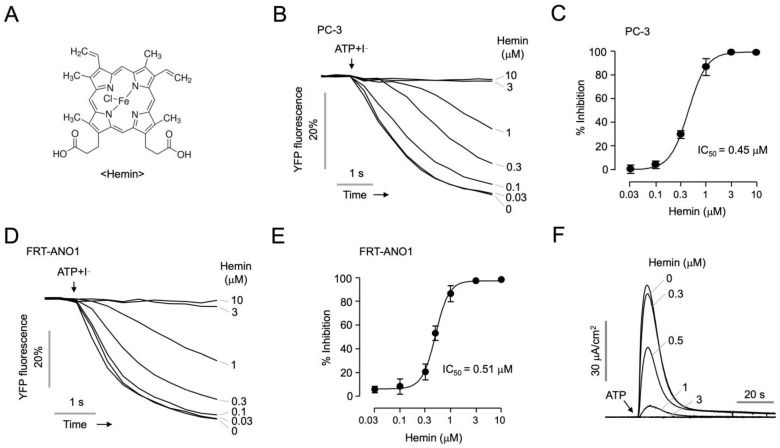
Identification of a novel ANO1 inhibitor, hemin. (**A**) Chemical structure of hemin. (**B**) The inhibitory effect of hemin was determined via YFP fluorescence quenching assay in PC-3 cells. PC-3 cells expressing mutant YFP (F46L/H148Q/I152L) were treated with the indicated concentrations of hemin for 10 min, and then ANO1 was activated by ATP (100 μM). (**C**) Dose–response curve; *n*_H_  =  2.4 (mean ± S.D.; *n* = 6). (**D**) The inhibitory effect of hemin on ANO1 channel activity was determined in FRT cells expressing human ANO1. The indicated concentrations of hemin were administered for 10 min prior to ATP (100 μM) treatment. (**E**) Dose–response curve on ANO1 inhibitory activity; *n*_H_  =  3.3 (mean ± S.D.; *n* = 5). (**F**) Apical membrane currents were measured in FRT cells expressing ANO1. The cells were pre-treated with the indicated concentrations of hemin for 10 min, and the ANO1 chloride currents were activated by ATP (100 μM).

**Figure 2 ijms-25-06032-f002:**
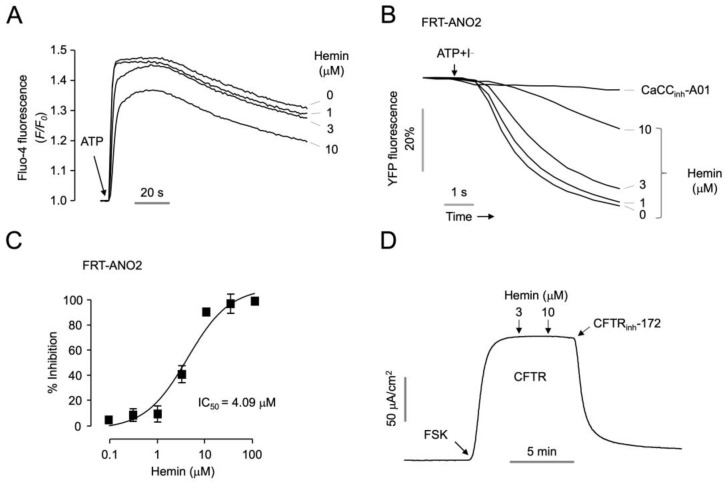
Characterization of the novel ANO1 inhibitor, hemin. (**A**) Intracellular calcium level was measured using Fluo-4 assay in PC-3 cells. The indicated concentration of hemin was treated for 10 min, and then ATP (100 μM) was applied. (**B**) Inhibitory effect of hemin on ANO2 determined by YFP fluorescence quenching assay. CaCC_inh_-A01 (10 µM) and the indicated concentrations of hemin were administered for 10 min prior to ATP (100 μM) treatment. (**C**) Dose–response curve on ANO2 inhibitory activity; *n*_H_  =  2.5 (mean ± S.D.; *n* = 5). (**D**) CFTR activity was measured in FRT cells expressing human CFTR. CFTR chloride currents were activated by 10 μM of forskolin (FSK), and then the indicated concentrations of hemin and CFTR_inh_-172 (10 μM) were applied.

**Figure 3 ijms-25-06032-f003:**
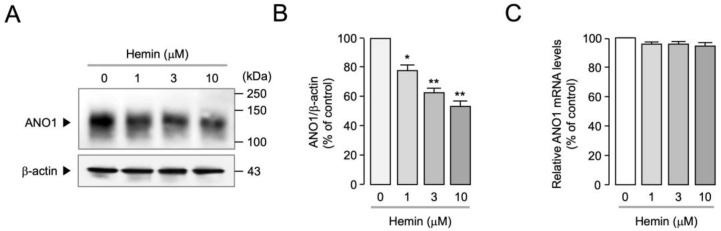
Effect of hemin on protein and mRNA expression levels of ANO1 in PC-3 cells. (**A**) PC-3 cells were treated with the indicated concentrations of hemin for 24 h, and then the expression levels of ANO1 and β-actin were analyzed via Western blot analysis. (**B**) ANO1 protein intensities were normalized to β-actin (mean ± S.E., *n* = 3). * *p* < 0.05, ** *p* < 0.01 vs. control. (**C**) PC-3 cells were treated with 1, 3, and 10 μM of hemin for 24 h, and ANO1 mRNA levels were measured via real-time PCR (mean ± S.E.; *n* = 3).

**Figure 4 ijms-25-06032-f004:**
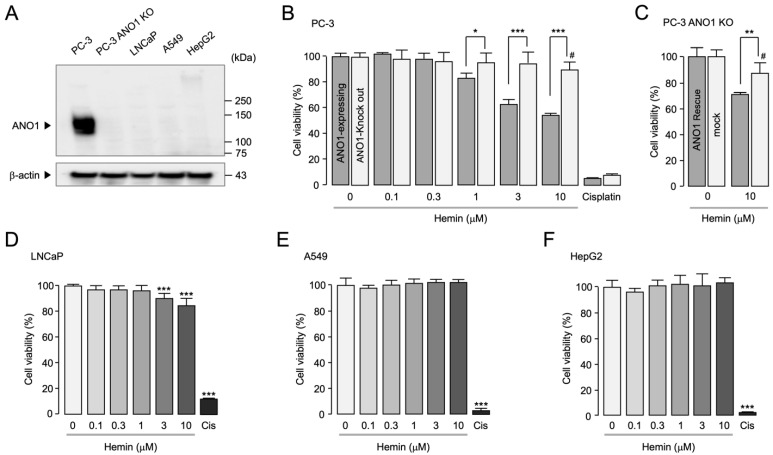
Expression levels of ANO1 protein and effect of hemin on cell viability in PC-3, LNCaP, A549, and HepG2 cells. (**A**) ANO1 protein expression levels in PC-3, PC-3 ANO1 knockout (KO), LNCaP, A549, and HepG2 cells. (**B**) Cell viability was assessed in PC-3 and PC-3 ANO1 KO cells. Cells were incubated with hemin at the indicated concentrations for 72 h, and medium was replaced every 24 h (mean ± S.D.; *n* = 5). * *p* < 0.05, *** *p* < 0.001; # *p* < 0.05 vs. ANO1 KO group control. (**C**) Cell viability was assessed in PC-3 ANO1 KO cells subjected to either mock transfection or ANO1 transfection. Cells were incubated with 10 μM of hemin for 72 h, and medium was replaced every 24 h (mean ± S.D.; *n* = 6). ** *p* < 0.01; # *p* < 0.05 vs. mock group control. (**D**–**F**) LNCaP, A549, and HepG2 cells were treated with hemin at the indicated concentrations for 72 h, and medium was changed every 24 h (mean ± S.D.; *n* = 5). Cisplatin was treated at 50 μM. *** *p* < 0.001 vs. control.

**Figure 5 ijms-25-06032-f005:**
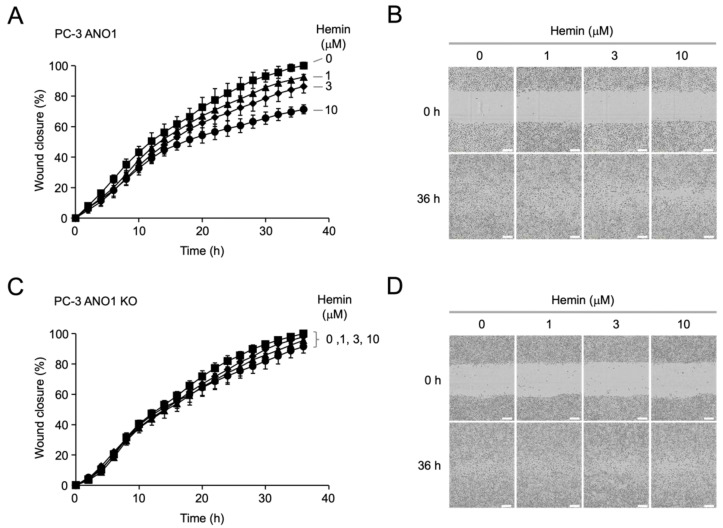
Effect of hemin on cell migration in PC-3 and PC-3 ANO1 KO cells. (**A**) Wound healing assay was performed in PC-3 cells. Cells were treated with the indicated concentrations of hemin, and wound closure was measured every 2 h after wound formation (mean ± S.E., *n* = 3). (**B**) Representative images were acquired at 0 and 36 h following administration of hemin at the indicated concentrations. (**C**) Wound healing assay was performed in PC-3 ANO1 KO cells. Cells were treated with the indicated concentrations of hemin, and wound closure was measured every 2 h after wound formation (mean ± S.E., *n* = 3). (**D**) Representative images were acquired at 0, 36 h following administration of hemin at the indicated concentrations. The scale bars represent 300 μm.

**Figure 6 ijms-25-06032-f006:**
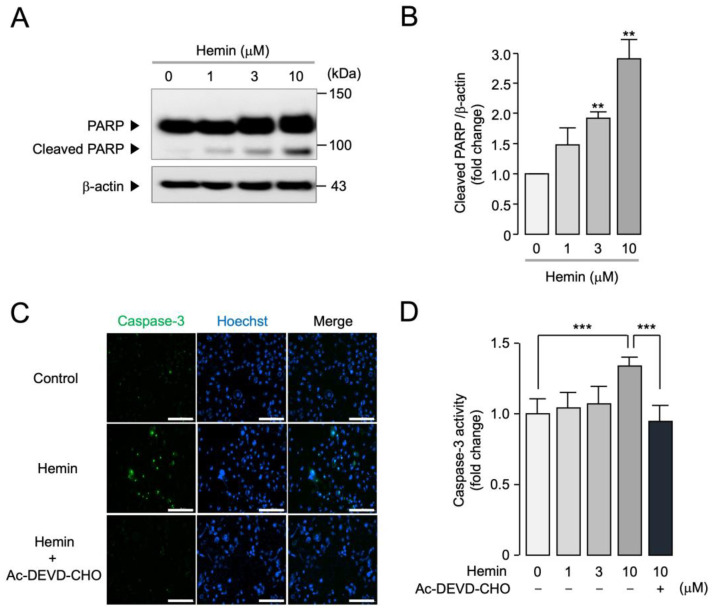
Effect of hemin on PARP cleavage and caspase-3 activity in PC-3 cells. (**A**) PC-3 cells were treated with 1, 3, 10 µM of hemin for 24 h, and then the expression levels of PARP, cleaved-PARP, and β-actin were analyzed via Western blot analysis. (**B**) Cleaved PARP intensity was normalized to β-actin (mean ± S.E., *n* = 3). ** *p* < 0.01 vs. control. (**C**) Images were taken 24 h after application of 10 µM of hemin in PC-3 cells, and then cells were incubated with 1 µM of caspase-3 substrate (green) and 1 µM of Hoechst 33342 (blue) for 30 min prior to image acquisition. Scale bars represent 200 µm. (**D**) PC-3 cells were treated with the indicated concentrations of hemin in the presence or absence of 20 µM of Ac-DEVD-CHO for 24 h and then treated with 1 µM of caspase-3 substrate for 30 min to estimate caspase-3 activity. *** *p* < 0.001.

**Figure 7 ijms-25-06032-f007:**
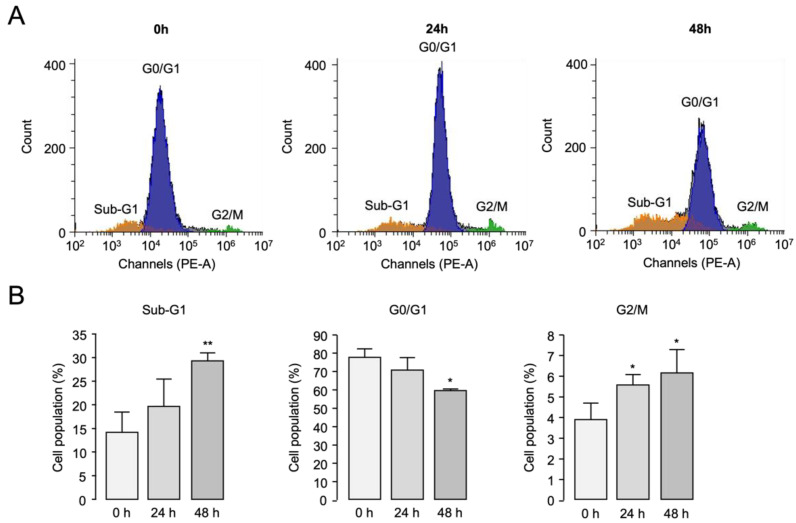
Effect of hemin on cell cycle in PC-3 cells. (**A**,**B**) Cell cycle phases were observed via propidium iodide (PI) staining followed by flow cytometric analysis after cells were treated with 3 μM of hemin for 24 h and 48 h (mean ± S.D.; *n* = 3). * *p* < 0.05, ** *p* < 0.01 vs. control.

## Data Availability

The data that support the findings of this study are available from the corresponding author upon reasonable request.

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
