# Peer review of "Anticancer Effect of Hemin through ANO1 Inhibition in Human Prostate Cancer Cells"

_ijms, 2024, doi:10.3390/ijms25116032_

Round 1

Reviewer 1 Report

Comments and Suggestions for Authors

see attachment

Author Response

We greatly appreciate the editor’s and reviewers’ efforts to carefully review our manuscript and the valuable comments and suggestions offered for the improvement of the manuscript (ijms-2989346). We have made each of the suggested revisions and the points of criticism raised by the reviewers were addressed by a point-by-point response. Changes in the manuscript text are highlighted in red color font.

Reviewer #1:

Comments to the Author

This work of Park et al. shows impressively, that hemin not only selectively inhibits ANO1 (and ANO2 to a weaker extent), but also decreases protein expression of ANO1. By using cell lines without endogenous ANO1 and PC-3 ANO1 KO cells, they revealed that hemin only effects the viability of PC-3 cells which endogenously express ANO1. Additional techniques like wound healing assay, detection of apoptosis hallmarks via WB and cell cycle analysis underline the validity of their data.

As the expression level of ANO1 is closely related to different types of cancer, this work is very important for the development of new forms of therapy. Nevertheless, I have a few small comments, questions or suggestions for improvement:

  1. Are the DRC’s done with a Hill-Fit? If so, what is the Hill-coefficient?

Response: The Hill-coefficient values were 2.4 for Figure 1C, 3.3 for Figure 1E, and 2.5 for Figure 2C. This information is described in the figure legends.

  1. If I understood correctly, in all experiments, the cells were incubated with hemin for at least 10 minutes. In Figure 2 D, there is acute application during the experiment with no effect. Did the authors also try this experiment with hemin-incubation?

Response: Thank you for the comments. In contrast to the transient activation of ANO1, a calcium-activated chloride channel, CFTR chloride channels exhibit a sustained activation profile. Therefore, when evaluating the effect of a compound on CFTR activity, CFTR is typically maximally activated using a high concentration of forskolin, and once a stable current is achieved, the test compound is introduced to assess its potential inhibitory impact on CFTR-mediated chloride currents. As shown in Figure 2D, treatment with a high concentration of hemin had no effect in CFTR channels, and complete inhibition was achieved by CFTRinh-172, a specific inhibitor of CFTR.

  1. As hemin is dissolved in DMSO, does the experiments with 0 μM hemin include DMSO (according to the highest concentration of hemin) or are there other control experiments to exclude any influence of the solvent?

Response: Thank you for the comments. 0 μM of hemin refers to the control containing DMSO alone. This information is described in Materials and Methods section.

  1. Did the authors also test for protein expression of ANO2, according to experiments shown in Fig.3?

Response: Thank you for the comments. We attempted to detect ANO2 protein expression by utilizing a commercially available antibody (sc-390956, Santa Cruz Biotechnology, Dallas, TX, USA), but were unable to obtain a reliable signal. However, when pretreated with Ani9, a specific inhibitor of ANO1 that does not affect ANO2, the ATP-induced calcium-activated chloride channel (CaCC) activity was almost completely inhibited in PC-3 cells. Therefore, functional ANO2 expression in PC-3 cells is expected to be minimal and not to interfere with the experiments conducted in this study.

  1. In line 145-148, the authors equal a decrease in cell proliferation and migration with cytotoxicity. In my opinion, this would be cytostatics. Cytotoxicity would lead to cell death. In general, a decrease in the number of viable cells may result from either cell death or reduced proliferation.
    Response: Thank you for the helpful comments. We agree with the reviewer's opinion. In the revised manuscript, we have modified the phrasing to clearly differentiate the observed effects on cell viability from cytotoxicity.

  1. Figure 4B: is the difference for ANO1-KO cells between application of 0 μM and 10 μM hemin significant? If yes, do the authors have an explanation for that?

Response: Thank you for your insightful comments. In PC-3 ANO1 knockout cells, no statistically significant differences were observed compared to the 0 μM hemin control at other hemin concentrations. However, in the group treated with 10 μM hemin, the P value was less than 0.05, indicating a statistically significant difference from the control. However, at 10 μM hemin, ANO1 KO PC-3 cells exhibited a slightly significant decrease in cell viability to 13.36%, whereas PC-3 cells showed a highly significant reduction to 45.74%. The observed effects of 10 μM hemin in PC-3 ANO1 knockout cells are likely due to off-target effects, such as on the heme oxygenase-1 (HO-1) enzyme. Additionally, as shown in Figure S2, treatment of PC-3 and PC-3 ANO1 knockout cells with 30 μM hemin strongly inhibited cell activity with no statistical difference in both cells. Therefore, high concentrations of hemin are thought to inhibit cell growth through alternative pathways and not only through regulation of ANO1. This information is described in the revised manuscript.

  1. The methods should be explained more detailed, e.g. the flow cytometry analysis of the cell cycle phases shown in Fig. 7 or the Caspase-3 activity assay.

Response: Thank you for the comments. More detailed information about analytical procedures, analytical methods, treatment conditions and time points has been revised in the Methods section.

  1. Figure 7: why did the authors choose a hemin concentration of 3 μM for these experiments?

Response: Thank you for the comments. In order to minimize potential off-target effects while ensuring complete inhibition of ANO1, we selected a concentration of 3 μM hemin for the flow cytometry experiments (Figure 7). This concentration was chosen based on our findings that hemin almost completely inhibits ANO1 activity at 3 μM (Figure 1). However, at 10 μM, hemin also demonstrated undesirable effects on intracellular calcium levels and ANO2 activity (Figure 2A, 2B). Therefore, we determined that 3 μM hemin provided a suitable balance, exhibiting robust inhibition of ANO1 while minimizing off-target effects on other targets. This information is described in the revised manuscript.

  1. In line 251, the authors state that data from another group revealed cytotoxicity for high concentrations of hemin (30-80 μM). Did the authors also try such high concentrations for the PC3 ANO1 KO cells?

Response: Thank you for the comments. As expected, there was no significant difference in cell viability between PC-3 and PC-3 ANO1 KO cells treated with 30 μM of hemin, which decreased to 49% and 40%, respectively (Figure S2). The loss of ANO1 selectivity at high hemin concentrations of hemin suggests that high concentrations of hemin may exhibit additional anticancer effects by other pathways such as increasing HO-1 expression. This information is described in the Discussion section.

  1. “In conclusion, hemin potently and selectively inhibited ANO1 channel activity and decreased the protein expression level of ANO1 without alteration of ANO1 mRNA expression in PC-3 cells.” As the inhibition of channel activation and protein expression are very different processes, is there any explanation why hemin influences both of them or is there any proposed mechanism for the two processes?

Response: Thank you for the comments. Previous studies have indicated that both ANO1 channel activity and ANO1 protein expression are associated with cancer progression. Suppressing ANO1 protein expression can not only decrease channel activity but also inhibit the cell growth signals mediated by ANO1 protein. Therefore, we hypothesize that pharmacological agents capable of reducing both ANO1 channel activity and protein levels may exhibit more potent anticancer effects. We have added more detailed information in the Discussion section.

  1. YFP fluorescence quenching assay: are the cells transiently transfected with YFP?

Response: In YFP fluorescence quenching assay, we used a stably cell line expressing YFP (F46L /H148Q/I152L). This information is described in Materials and Methods section.

  1. Supplementary Figure 6: what are the bands below “cleaved PARP”?

Response: Thank you for the comments. The band observed below cleaved PARP is considered a non-specific signal, as its intensity exhibits a marked decline upon hemin treatment, even at 1 μM. This pattern is distinct from the anticipated dose-dependent increase in cleaved PARP levels induced by hemin, suggesting that this band likely corresponds to an unrelated protein whose expression is downregulated by hemin exposure.

Minor:

  1. Line 272: where are the cell lines from?

Response: Thank you. Corrected.

  1. Line 283: “4.3. YFP Fluorescnece Quenching Assay“ – Fluorescence

Response: Thank you. Corrected.

  1. Full blots: marker is missing, or at least indication of the size of the bands

Response: Thank you. Corrected.

Reviewer 2 Report

Comments and Suggestions for Authors

In this manuscript, So-Hyeon Park and collaborators investigated the role of Hemin/ANO1 axis on prostate carcinoma cells. Although hemin had no significant effect on intracellular calcium signaling, ANO1 expression was significantly reduced in hemin-treated cell and cell proliferation and migration was impaired.

Comments/Suggestions:

Overall, the manuscript is well organized and the conclusions are supported by the results.

 1)     Most of the experiments were conducted in PC-3 cells line. To justify the title of the manuscript, the authors should confirm the role of ANO1 in another prostate cancer cell line expressing ANO1.

2)     Fig 4: Rescue experiments should be performed in PC-3 ANO1 knockout cells to demonstrate the specific impact of ANO1 on prostate cancer cell proliferation and migration.

3)     Fig 5: Did the authors use ARA-C to exclude the contribution of proliferation on migration assay?

Comments on the Quality of English Language

 Minor editing of English language is required

Author Response

We greatly appreciate the editor’s and reviewers’ efforts to carefully review our manuscript and the valuable comments and suggestions offered for the improvement of the manuscript (ijms-2989346). We have made each of the suggested revisions and the points of criticism raised by the reviewers were addressed by a point-by-point response. Changes in the manuscript text are highlighted in red color font.

Reviewer #2:

In this manuscript, So-Hyeon Park and collaborators investigated the role of Hemin/ANO1 axis on prostate carcinoma cells. Although hemin had no significant effect on intracellular calcium signaling, ANO1 expression was significantly reduced in hemin-treated cell and cell proliferation and migration was impaired.

Comments/Suggestions:

Overall, the manuscript is well organized and the conclusions are supported by the results.

  1. Most of the experiments were conducted in PC-3 cells line. To justify the title of the manuscript, the authors should confirm the role of ANO1 in another prostate cancer cell line expressing.

Response: Thank you for the comments. Among the various prostate cancer cell lines, PC-3 cells exhibit the highest endogenous expression levels of the ANO1 protein (Cell Death & Disease. 2018 Jun; 9(6): 703; Figure 1). Indeed, we were unable to detect functional expression of ANO1 in DU145 and LNCaP prostate cancer cell lines. So, all experiments were conducted using the PC-3 cells robustly expressing ANO1.

  1. Fig 4: Rescue experiments should be performed in PC-3 ANO1 knockout cells to demonstrate the specific impact of ANO1 on prostate cancer cell proliferation and migration.

Response: Thank you for your helpful comments. We examined the effect of ANO1 rescue in PC-3 ANO1 KO cells on hemin-induced cytotoxic effect (Figure 4C). In PC-3 ANO1 KO cells transfected with ANO1, cell viability was significantly reduced to 29.28% following treatment with 10 μM hemin compared to the control group. This information is described in the revised manuscript.

  1. Fig 5: Did the authors use ARA-C to exclude the contribution of proliferation on migration assay?

Response: Thank you for the comments. We did not use ARA-C since the ETS-related gene (ERG) is overexpressed in PC-3 cells (BMC Cancer. 2015 Aug 27:15:604) and ARA-C is known to inhibit ERG. Instead, to minimize the contribution of proliferative effects, we performed the cell migration assays in growth medium supplemented with a reduced concentration of 2% fetal bovine serum (FBS). This information is described in the Materials and Methods section.

Reviewer 3 Report

Comments and Suggestions for Authors

 So-Hyeon Park et al. investigated the role of hemin in inhibiting prostate cancer through ANO1 Inhibition in vitro. They showed that hemin is a selective ANO1 inhibitor and its anticancer effect in PC-3 cells depends on ANO1 expression. The research is interesting and the publication should be slightly revised.

The authors write that "hemin induced a decrease in the G0/G1 phase ..... and an increase in the G2/M phase ...... These results indicated that hemin exerted an anticancer effect through apoptosis." The authors should consider the role of cell cycle inhibition in individual phases and complete the final conclusion.

In the Discussion section, the authors should be careful in interpreting the antitumor effects of hemin based on the effect on HO-1 activity. It is known that both inhibition and induction of HO-1 can exert an antitumor effect in prostate cancer models and this effect may depend on the intensity of enzymatic activity [see review: Curr. Issues Mol. Biol. 2023, 45(5), 4301-4316].

There are too many self-citations in the publication.

Author Response

We greatly appreciate the editor’s and reviewers’ efforts to carefully review our manuscript and the valuable comments and suggestions offered for the improvement of the manuscript (ijms-2989346). We have made each of the suggested revisions and the points of criticism raised by the reviewers were addressed by a point-by-point response. Changes in the manuscript text are highlighted in red color font.

Reviewer #3:

So-Hyeon Park et al. investigated the role of hemin in inhibiting prostate cancer through ANO1 Inhibition in vitro. They showed that hemin is a selective ANO1 inhibitor and its anticancer effect in PC-3 cells depends on ANO1 expression. The research is interesting and the publication should be slightly revised.

  1. The authors write that "hemin induced a decrease in the G0/G1 phase ..... and an increase in the G2/M phase ...... These results indicated that hemin exerted an anticancer effect through apoptosis." The authors should consider the role of cell cycle inhibition in individual phases and complete the final conclusion.

Response: Thank you for your insightful comments. We have updated the final conclusion to reflect the detailed findings, emphasizing the role of hemin in inducing apoptosis and causing cell cycle arrest.

  1. In the Discussion section, the authors should be careful in interpreting the antitumor effects of hemin based on the effect on HO-1 activity. It is known that both inhibition and induction of HO-1 can exert an antitumor effect in prostate cancer models and this effect may depend on the intensity of enzymatic activity [see review: Curr. Issues Mol. Biol. 2023, 45(5), 4301-4316].

Response: Thank you for your thoughtful comments. In the discussion section, we described the dual role of HO-1 in prostate cancer.

  1. There are too many self-citations in the publication.

Response: Thank you for your comments. Because our research team conducted research to develop and evaluate the activity of various ANO1 inhibitors, we unintentionally self-cited a lot. Efforts were made to reduce self-citations without damaging the content of the manuscript, and ultimately the number of self-citations was adjusted to 8 out of 46 (17.39%).

Round 2

Reviewer 2 Report

Comments and Suggestions for Authors

All the raised questions have been addressed. The revised version of the manuscript is improved.